Iranian and Swedish adolescents: differences in personality traits and well-being

Kjell Oscar N.E. 1 oscar.kjell@psy.lu.se
Nima Ali A. 2 3
Sikström Sverker 1
Archer Trevor 2 3
Garcia Danilo 3 4 5 danilo.garcia@neuro.gu.se danilo.garcia@euromail.se
1 Department of Psychology, Lund University , Lund , Sweden
2 Department of Psychology, University of Gothenburg , Gothenburg , Sweden
3 Network for Empowerment and Well-Being , Sweden
4 Center for Ethics, Law and Mental Health (CELAM), University of Gothenburg , Gothenburg , Sweden
5 Institute of Neuroscience and Physiology, The Sahlgrenska Academy, University of Gothenburg , Gothenburg , Sweden
Moss Timothy
Electronic publication date: 2013 Nov 5
Publication date: 2013
Volume: 1
Electronic Location ID: e197
Received 2013 Sep 5; Accepted 2013 Oct 10
Copyright: © 2013 Kjell et al.
Copyright year: 2013
Copyright holder: Kjell et al.
License: This is an open access article distributed under the terms of the Creative Commons Attribution License, which permits unrestricted use, distribution, and reproduction in any medium, provided the original author and source are credited.
License URL: https://creativecommons.org/licenses/by/3.0/

Keywords: Personality traits, Big Five, Subjective well-being, Psychological well-being, Adolescence, Cross-cultural, Iran, Sweden

Funding: Stiftelsen Kempe-Carlgrenska Fonden This study was supported by Stiftelsen Kempe-Carlgrenska Fonden. The funder had no role in study design, data collection and analysis, decision to publish, or preparation of the manuscript.

==============================
Introduction. This study addresses the need to further contextualize research on well-being (e.g., Kjell, 2011) in terms of cross-cultural aspects of personality traits among adolescents and by examining two different conceptualizations of well-being: subjective well-being (i.e., life satisfaction, positive and negative affect) and psychological well-being (i.e., positive relations with others, environmental mastery, self-acceptance, autonomy, personal growth, and life purpose).

Methods. Iranian (N = 122, mean age 15.23 years) and Swedish (N = 109, mean age 16.69 years) adolescents were asked to fill out a Big Five personality test, as well as questionnaires assessing subjective well-being and psychological well-being.

Results. Swedes reported higher subjective and psychological well-being, while Iranians reported higher degree of Agreeableness, Openness and Conscientiousness. Neuroticism and Extraversion did not differ between cultures. Neuroticism was related to well-being within both cultures. Openness was related to well-being only among Iranians, and Extraversion only among Swedes. A mediation analysis within the Swedish sample, the only sample meeting statistical criteria for mediation analysis to be conducted, demonstrated that psychological well-being mediated the relationship between Neuroticism and subjective well-being as well as between Extraversion and subjective well-being.

Conclusions. Certain personality traits, such as Extraversion, Openness, and Conscientiousness, relate differently to well-being measures across cultures. Meanwhile, Neuroticism seems to relate similarly across cultures at least with regard to subjective well-being. Furthermore, the results give an indication on how psychological well-being might mediate the relationship between certain personality traits and subjective well-being. Overall, the complexity of the results illustrates the need for more research whilst supporting the importance of contextualizing well-being research.

The study of well-being during adolescence is important since this period of life is characterized by various events and transitions that significantly influence adolescents’ well-being (González, Casas & Coenders, 2007). Although research on adolescents’ well-being has gained terrain in the last decade (e.g., see Garcia & Archer, 2012; Garcia & Siddiqui, 2009a; Garcia & Siddiqui, 2009b; Garcia & Sikström, 2013; Fogle, Huebner & Laughlin, 2002; Funk, Huebner & Valois, 2006), the study of well-being across cultures is still scarce (for a recent review showing that the majority of previous research in this area involves American participants see Proctor, Linley & Maltby, 2009). Cross-cultural research is important because indices of well-being behave differently in cross-cultural comparative research (Beirens & Fontaine, 2011; Diener et al., 2009). Herein, cross-cultural differences between adolescents from Sweden and Iran are examined in relation to differences in personality traits across cultures. Within this framework the relationships between subjective well-being and psychological well-being are also explored.

Swedes, as compared with Iranians, tend to report higher levels of well-being in several global polls (e.g., Veenhoven, 2013; Diener & Tov, 2009). Notably, though, these comparisons most often involve measures of subjective well-being (e.g., happiness and life satisfaction) in relation to important societal variables (e.g., crime, education and health) (Veenhoven, 2012) rather than cross-cultural variations in psychological well-being or personality measures. Importantly though, research has recently demonstrated the increasing importance of personality traits and its diverse and strong influences on various aspects of well-being (Lucas & Diener, 2008; Kim-Prieto et al., 2005; Steel, Schmidt & Shultz, 2008). Further, in relation to global polls, personality research shows that cultures that are geographically far from each other tend to exhibit differences in personality traits (Allik & McCrae, 2004). There are clear differences between European and American cultures as compared with Asian and African cultures: Europeans and Americans, for instance, report being higher in Extraversion and Openness and lower in Agreeableness (Allik & McCrae, 2004). How these cross-cultural personality differences relate to well-being is important — especially considering that personality is one of the most important determinants of well-being (Kim-Prieto et al., 2005). Hence, since Iran and Sweden are geographically far away from each other, being positioned in Europe versus Asia respectively, we expect their personality profiles to be different. Further, this expected variation might address the question specifically under investigation herein: are different personality traits related to different well-being conceptions across cultures?

Subjective Well-being and Psychological Well-Being

In the current study two distinct approaches were employed to comprehensively capture the concept of well-being (for recent discussions on the benefits of employing a variety of well-being measures see: Biswas-Diener, Kashdan & King, 2009; Delle Fave & Bassi, 2009; Garcia, 2013; Garcia et al., 2013a; Garcia et al., 2013b; Kashdan, Biswas-Diener & King, 2008; Ryan & Huta, 2009; Straume & Vittersø, 2012; Waterman, 2008). The first approach, subjective well-being (Diener, 1984), conceptualizes high well-being as the assessment of individuals’ own judgments of high life satisfaction, high frequency of positive affect and low frequency of negative affect. The second approach, psychological well-being (Ryff, 1989), includes 6 distinct dimensions involving: positive relations with others, environmental mastery, self-acceptance, autonomy, personal growth, and life purpose. These 6 dimensions define psychological well-being both theoretically and operationally, and they identify what promotes effective mastery of life events and emotional and physical health (Ryff, 1989). Although both approaches can be seen to reflect Western cultures (e.g., Christopher, 1999); the two different ways to measure well-being differ in that the psychological well-being approach consist of predefined criteria (i.e., the 6 dimensions) meanwhile the subjective well-being approach is comparatively more ‘open’ in allowing the respondents to decide the criteria for themselves (although within the predefined criteria of satisfaction). Psychological well-being is often theorized to promote subjective well-being (Ryan & Deci, 2001). For example, among Swedish adolescents, psychological well-being, and especially the subscale of self-acceptance, is strongly related to subjective well-being (Garcia, 2011a; Garcia, 2011b; Garcia, 2012a; Garcia & Archer, 2012; Garcia & Siddiqui, 2009b). This relationship is significant even when personality traits are controlled for. The current study will expand these findings by examining if psychological well-being mediates the relationship between certain personality traits and subjective well-being.

Personality Traits and Well-Being

In the present study, personality traits are operationalized employing the Big Five Inventory (Benet-Martínez & John, 1998). This is a valid and reliable instrument measuring five dimensions: Extraversion, Neuroticism, Agreeableness, Conscientiousness and Openness (for a review see, John, Naumann & Soto, 2008). Neuroticism and Extraversion appear to be most significant in predicting adults’ and adolescents’ subjective well-being; where Extraversion tends to correlate positively and Neuroticism negatively with subjective well-being (e.g., Fogle, Huebner & Laughlin, 2002; Lucas, 2008). Extraversion seems to influence subjective well-being because it is positively related to positive affect and being more attentive to positive experiences; whilst Neuroticism appears to be strongly negatively related to negative affect as well as being more prone to react more intensely to negative experiences (Larsen & Eid, 2008). However, Vittersø (2001) and DeNeve & Cooper (1998) put forward evidence that Extraversion is overrated as a predictor of subjective well-being. For example, Vittersø (2001) demonstrated that Neuroticism not only predicts the presence of negative affect but also the absence of positive affect better than Extraversion. Furthermore, while the influence of Neuroticism on subjective well-being appears analogous for adolescents and adults (e.g., Fogle, Huebner & Laughlin, 2002), recent research on adolescents yield mixed results for the trait of Extraversion. For instance, Rigby & Huebner (2005) suggest that specific avoidant behavior (e.g., avoiding standing out) in some adolescents might reduce the advantages of Extraversion that is seen among adults (see also Garcia, 2011a). Thus, meanwhile the negative relationship between Neuroticism and subjective well-being appears rather straightforward; the proposed positive relationship between Extraversion and subjective well-being seems more complex.

Agreeableness and Conscientiousness yield small to moderate positive correlations with measures of subjective well-being; whilst Openness shows no clear correlation with subjective well-being (DeNeve & Cooper, 1998; Diener & Seligman, 2002; Lucas, 2008). Nevertheless, findings by Allik & McCrae (2004) show that certain traits are more (un)common in certain cultures compared to others. The saliency or relative lack of these traits should serve a specific function — for example; these specific traits could be related to well-being within that culture.

The discussed patterns between personality traits and subjective well-being appear to overlap with the research on psychological well-being, wherein, among adults, high levels of Extraversion and Conscientiousness, along with low levels of Neuroticism predict high levels of psychological well-being (Ryff, Keyes & Shmotkin, 2002). These specific findings have also been replicated among Swedish adolescents (Garcia, 2011a).

The Present Study

The current study addresses the need to further contextualize research on well-being (e.g., see Kjell, 2011) in terms of cross-cultural aspects of personality traits among adolescents. We begin to draw two hypotheses that are rather straightforward in the empirical literature, that is, we expect: (i) personality traits to vary between nationalities and (ii) Swedish, as compared with Iranian, adolescents to report higher scores on most dimensions of well-being. However, since current research on the relationship between well-being and personality traits is, to date, somewhat scarce in terms of cross-cultural variations, we are more careful in terms of these hypotheses. We expect (iii) personality traits to be related to the two current well-being measures, so that:

(a) Considering the strong evidence that Neuroticism consistently is negatively related to well-being measures, both Swedish and Iranian data are hypothesized to demonstrate this negative relationship. However, if these cultures show differences in other personality traits; these traits are also expected to be related to well-being.

(b) Psychological well-being will be related to subjective well-being as well as mediating the relationship between personality traits and subjective well-being.

Method

Participants and procedure

The data was collected at a high school south of Sweden (N = 109, 38 boys and 71 girls, mean age 16.69 years SD .91 years) and two high schools in Tehran and Zanjan, Iran (N = 122, 72 boys and 50 girls, mean age 15.23 years SD 1.26 years). The sampling procedure of schools was based on convenience and included the entire schools; although they had no particular interest or knowledge about our research interest beforehand. Teachers and parents were informed about the nature of the study and that participation was voluntary and that pupils had the right to withdraw at any moment. The school nurse from each school was contacted by the researchers and informed about the study in case any of the students needed counseling. Participants were informed that the study examined how pupils think about their lives in different situations. They were ensured anonymity and informed that participation was voluntary; they had consent from their teachers to participate. The study was conducted in the participants’ own classrooms in groups of 20 to 30 pupils; the questionnaires were distributed on paper. The entire procedure, including debriefing, took approximately 30 min. The Ethics Committee of Gothenburg University approved this research protocol and written informed consent was obtained from all the participants.

Questionnaires

The Big Five Inventory (Benet-Martínez & John, 1998) is a 44-item (5-point Likert scale: 1 = strongly disagree to 5 = strongly agree) instrument that enables efficient assessment of the five personality dimensions: Neuroticism (e.g., I see myself as a person “who worries a lot”), Extraversion (e.g., I see myself as a person “who is talkative”), Openness (e.g., I see myself as a person who “is original, has new ideas”), Agreeableness (e.g., I see myself as a person who “has a forgiving nature”), and Conscientiousness (e.g., I see myself as a person who “does things efficiently”). This self-report measure has been empirically demonstrated to be apt for cross-language and cross-cultural research (Benet-Martínez & John, 1998; Allik & McCrae, 2004). For the Swedish sample the instrument was translated from English to Swedish and then backtranslated by Swedish native speakers; no significant discrepancies were found. Cronbach’s α varied between .84 and .92 among traits. The Iranian version has been used in previous studies (e.g., Joshanloo & Afshari, 2009; Joshanloo & Nosratabadi, 2009). For the Iranian version Cronbach’s α varied between .65 and .72 in the present study.

The Satisfaction With Life Scale (Diener et al., 1985) assesses the cognitive component of subjective well-being (i.e., life satisfaction) and consists of 5 items (e.g., “In most of my ways my life is close to my ideal”) that require a response on a 7-point Likert scale (1 = strongly disagree, 7 = strongly agree). Both the Swedish and the Iranian versions of this instrument have previously been used in these cultures (e.g., Garcia, 2011c; Garcia, 2012a; Garcia, 2012b; Garcia, Rosenberg & Siddiqui, 2011; Garcia & Moradi, 2012; Garcia & Moradi, 2013; Jokar, Samani & Sahragard, 2007; Joshanloo & Ghaedi, 2009). In the current study the instrument had a Cronbach’s α = .83 in the Swedish sample and .87 in the Iranian sample.

The Positive Affect and Negative Affect Schedule (Watson, Clark & Tellegen, 1988) assesses the affective components of subjective well-being by requiring participants to indicate on 5-point Likert scale to what extent (1 = very slightly, 5 = extremely) they generally experienced 20 different adjectives (10 positive affect and 10 negative affect) within the last few weeks. The positive affect scale includes adjectives such as strong, proud, and interested; and the negative affect scale includes adjectives such as afraid, ashamed, and nervous. The Swedish version has been used in previous studies (e.g., Garcia et al., 2012a; Garcia et al., 2012b; Garcia, Kerekes & Archer, 2012; Garcia & Erlandsson, 2011; Garcia et al., 2010; Nima, Archer & Garcia, 2012; Nima et al., 2013; Schütz et al., 2013) and demonstrated good internal consistency in the present study (positive affect Cronbach’s α = .86, negative affect Cronbach’s α = .78). The Iranian version was translated from English to Farsi and backtranslated by Farsi native speakers; no significant discrepancies were found. The Iranian version demonstrated low but acceptable internal consistence; Cronbach’s α = .56 for positive affect and .57 for negative affect.

The Scales of Psychological Well-Being (the short version; Clarke et al., 2001) comprises 18 items, 3 items for each of the 6 dimensions, using a 6-point Likert scale (1 = strongly disagree, 6 = strongly agree). These dimensions are: (i) positive relations with others (e.g., “People would describe me as a giving person, willing to share my time with others”), (ii) environmental mastery (e.g., “I am quite good at managing the responsibilities of my daily life”), (iii) self-acceptance (e.g., “I like most aspects of my personality”), (iv) autonomy (e.g., “I have confidence in my own opinions, even if they are contrary to the general consensus”), (v) personal growth (e.g., “For me, life has been a continuous process of learning, changing, and growth”), and (vi) purpose in life (“Some people wander aimlessly through life, but I am not one of them”). The Swedish version has been used in previous studies (e.g., Garcia, 2011a; Garcia & Siddiqui, 2009b; Nima et al., 2013) and in the current study the total psychological well-being score (i.e., the sum of the 18 items) demonstrated a Cronbach’s α of .72. The Iranian version has also been used in several studies (e.g., Garcia & Moradi, 2013; Joshanloo & Ghaedi, 2009) and in the current study the total psychological well-being score demonstrated a Cronbach’s α of .62.

Results

The initial analyses examine differences in personality traits, subjective well-being and psychological well-being between the national samples by employing three Multivariate Analyses of Variance (MANOVA). Thereafter the relationship between personality as a predictor of well-being is examined by means of regression analyses within each sample; which further allows for the use of mediational analyses.

Differences between Swedish and Iranian adolescents

Table 1 shows mean scores and standard deviations as well as indicating significant differences for all variables in both samples. Independent factors for all MANOVAs were nationality and gender. For the first MANOVA the dependent factors were the Big Five personality traits, for the second, the three components of subjective well-being, and for the third, the six dimensions of psychological well-being and its total score.

Table 1 Mean scores for all variables in the study in both samples.

	Swedish boys	Swedish girls	Swedish total	Iranian boys	Iranian girls	Iranian total	Theoretical
range	
Extraversion	3.48 ± .29	3.48 ± .46	3.49 ± .41	3.37 ± .54	3.55 ± .47	3.44 ± .51	1–5	
Neuroticism	2.69 ± .50c ***	3.24 ± .42b ***	3.05 ± .52	3.03 ± .53	3.12 ± .57	3.06 ± .54	1–5	
Agreeableness	3.08 ± .40	3.48 ± .38b *	3.34 ± .43	3.51 ± 1.02	3.59 ± .51	3.54 ± .85**	1–5	
Conscientiousness	3.22 ± .58	3.14 ± .50	3.17 ± .52	3.49 ± .48	3.42 ± .57	3.46 ± .52***	1–5	
Openness	3.03 ± .32	3.36 ± .37b *	3.25 ± .39	3.83 ± .60	3.87 ± .79	3.85 ± .68***	1–5	
Life satisfaction	5.14 ± 1.07	4.67 ± 1.26	4.82 ± 1.22	4.53 ± 1.72	4.87 ± 1.59	4.66 ± 1.67	1–7	
Positive affect	3.74 ± .67a **	3.33 ± .71	3.46 ± .72***	2.77 ± .60c ***	3.59 ± .82	3.08 ± .80	1–5	
Negative affect	2.02 ± .54	2.50 ± .64	2.34 ± .65	2.93 ± .39d **	2.49 ± .80	2.79 ± .62***	1–5	
Psychological well-being	4.49 ± .51d **	4.20 ± .49	4.29 ± .51*	4.14 ± .52	4.23 ± .60	4.17 ± .55	1–6	
Positive relations	4.26 ± .86	4.52 ± .86b **	4.44 ± .86***	3.65 ± 1.27	4.10 ± .86b **	3.82 ± 1.15	1–6	
Environmental mastery	4.57 ± .75a ***	3.96 ± .91	4.15 ± .90	4.56 ± .95a ***	4.07 ± 1.12	4.38 ± 1.04	1–6	
Self-acceptance	4.81 ± .79d ***	4.11 ± 1.07	4.33 ± 1.04	4.19 ± .75	4.63 ± 1.17b *	4.36 ± .95	1–6	
Autonomy	4.18 ± .81a *	3.84 ± .71	3.95 ± .75	4.79 ± .95a *	4.61 ± 1.00	4.72 ± .97***	1–6	
Personal growth	4.67 ± .91	4.52 ± .80	4.57 ± .83***	3.93 ± 1.00	3.63 ± .79	3.82 ± .94	1–6	
Purpose in life	4.46 ± .95	4.25 ± .72	4.32 ± .80***	3.74 ± .95	4.32 ± 1.12b **	3.95 ± 1.05	1–6	
Notes.

* p < .05.

** p < .01.

*** p < .001.

a Higher than girls.

b Higher than boys.

c Lower than all groups.

d Higher than all groups.

The first MANOVA showed that nationality (F(5, 192) = 15.27, p < .001, Wilks’ Lambda = .72, Observed Power = 1.00) and gender (F(5, 192) = 6.05, p < .001, Wilks’ Lambda = .86, Observed Power = 1.00) had an effect on the personality traits. The interaction of nationality and gender was also significant (F(5, 192) = 2.91, p < .01, Wilks’ Lambda = .93, Observed Power = .84). Iranian adolescents scored significantly higher in Agreeableness (F(1, 196) = 6.41, p < .01, Observed Power = .71), Openness (F(1, 196) = 18.89, p < .001, Observed Power = 1.00) and Conscientiousness (F(1, 196) = 12.17, p < .001, Observed Power = .94). Overall, the national samples did not differ significantly in Extraversion and Neuroticism; however, Swedish boys scored significantly lowest and Swedish girls scored significantly higher than boys. In Neuroticism (F(1, 196) = 9.03, p < .001, Observed Power = .99). See Table 1 for more details.

The second MANOVA showed that nationality (F(3, 201) = 12.23, p < .001, Wilks’ Lambda = .85, Observed Power = 1.00) had a significant effect on the subjective well-being measures. Although the effect of gender was not significant (p = .20), the interaction of nationality and gender was significant (F(3, 201) = 20.48, p < .001, Wilks’ Lambda = .77, Observed Power = 1.00). Swedish adolescents reported higher positive affect (F(1, 203) = 12.02, p < .001, Observed Power = .93) and lower negative affect (F(1, 203) = 26.29, p < .001, Observed Power = 1.00) than Iranian adolescents. Of all the groups, Iranian boys reported lowest positive affect and highest negative affect (see Table 1). No difference in life satisfaction was found between cultures and the interaction of nationality and gender had no effect on life satisfaction either.

The third MANOVA showed that nationality (F(6, 191) = 14.24, p < .001, Wilks’ Lambda = .69, Observed Power = 1.00) and gender (F(1, 191) = 6.21, p < .001, Wilks’ Lambda = .84, Observed Power = 1.00) had a significant effect on the psychological well-being measures. The interaction of nationality and gender was also significant (F(6, 191) = 4.37, p < .001, Wilks’ Lambda = .88, Observed Power = 1.00). Swedish adolescents scored higher on positive relations (F(1, 196) = 11.79, p < .001, Observed Power = .93), personal growth (F(1, 196) = 38.35, p < .001, Observed Power = 1.00), purpose in life (F(1, 196) = 5.87, p < .001, Observed Power = .67), and the psychological well-being total score (F(1, 196) = 4.28, p < .05, Observed Power = .54). Iranian adolescents scored higher on autonomy (F(1, 196) = 28.32, p < .001, Observed Power = 1.00). Moreover, boys as a group scored higher on environmental mastery (F(1, 196) = 15.36, p < .001, Observed Power = .97). The interaction of nationality and gender was significant for the psychological well-being total score (F(1, 196) = 5.65, p < .01, Observed Power = .66) and self-acceptance (F(1, 196) = 15.83, p < .001, Observed Power = .98). Specifically, Swedish boys reported higher scores on both psychological well-being total score and self-acceptance. Iranian girls reported higher self-acceptance than Iranian boys (see Table 1).

Relationships between personality and well-being within samples

Swedish Adolescents:Table 2 displays the correlations between personality and well-being variables for the Swedish sample. Multiple Regression Analysis (MRA) showed that Extraversion was positively related to life satisfaction; Neuroticism was negatively related to life satisfaction and positive affect and positively related to negative affect (see Table 3). Neuroticism, Extraversion, and Conscientiousness were associated to the psychological well-being total score. Among the six psychological well-being dimensions, Neuroticism was negatively associated and Extraversion was positively associated to positive relations (see also Agreeableness which was positively associated to this psychological well-being dimension), environmental mastery (see also Conscientiousness which was positively associated to this psychological well-being dimension), self-acceptance, and autonomy. Personal growth was positively associated to the traits of Conscientiousness and Openness, while purpose in life was solely positively associated to Conscientiousness (see Table 2).

Table 2 Correlations between personality and well-being variables in the Swedish sample.

	1	2	3	4	5	6	7	8	9	10	11	12	13	14	15	
Extraversion (1)	–															
Neuroticism (2)	−.35***	–														
Agreeableness (3)	.14ns	.17ns	–													
Conscientiousness (4)	.25**	−.26**	.03ns	–												
Openness (5)	.08ns	.25**	.38***	−.08ns	–											
Life satisfaction (6)	.52***	−.49***	.11ns	.21*	−.09ns	–										
Positive affect (7)	.31***	−.44***	−.18ns	.27**	−.17ns	.45***	–									
Negative affect (8)	−.23**	.49***	−.02ns	−.18ns	.21*	−.38***	−.32***	–								
Psychological Well-Being (9)	.53***	−.63***	.03ns	.52***	−.01ns	.67***	.38***	−.38***	–							
Positive relations (10)	.49***	−.33***	.38***	.13ns	.20ns	.42***	.05ns	−.19ns	.57***	–						
Environmental mastery (11)	.44***	−.65***	−.11ns	.42***	−.18ns	.49***	.51***	−.51***	.72***	.37***	–					
Self-acceptance (12)	.53***	−.54***	.03ns	.27**	−.10ns	.71***	.38***	−.29**	.76***	.46***	.52***	–				
Autonomy (13)	−.07ns	−.44***	−.16ns	.19ns	−.04ns	.15ns	.06ns	−.11ns	.36***	−.03ns	.17ns	.06ns	–			
Personal growth (14)	.32***	−.12ns	.16ns	.35***	.31***	.29**	.15ns	−.07ns	.60***	.18ns	.17ns	.30***	.17ns	–		
Purpose in life (15)	.05ns	−.09ns	−.22*	.50***	−.21*	.17ns	.13ns	−.12ns	.45***	−.07ns	.21*	.17ns	.02ns	.33***	–	
Notes.

ns nonsignificant

* p < .05.

** p < .01.

*** p < .001.

Table 3 Multiple regressions for the Swedish sample.

Personality traits’ relationship to subjective well-being and psychological well-being, as well as the relationship between psychological well-being dimensions and subjective well-being.

Predictor variable	Outcome variable	Adj R2	Unst. B	Unst. SE	Stand. β	F	t	
Personality traits	
Extraversion (E)		–	1.06	.33	.34	–	3.23***	
Neuroticism (N)	Life satisfaction	–	−.91	.26	−.38	–	−3.47***	
E, N		.33	–	–	–	8.22***	–	
Neuroticism	Positive affect	.21	−.45	.17	−.32	4.82	−2.63**	
Neuroticism	Negative affect	.23	.65	.15	.53	5.23	4.30***	
Extraversion		–	.23	.11	.18	–	2.10*	
Neuroticism	Psychological	–	−.52	.09	−.53	–	−5.81***	
Conscientiousness (C)	Well-Being	–	.32	.08	.33	–	4.00***	
E, N, C		.53	–	–	–	18.50***	–	
Extraversion		–	.86	.21	.41	–	4.10***	
Neuroticism	Positive relations	–	−.41	.17	−.25	–	−2.40*	
Agreeableness (A)		–	.73	.20	.36	–	3.65***	
E, N, A		.39	–	–	–	11.02***	–	
Extraversion		–	.41	.19	.19	–	2.14*	
Neuroticism	Environmental mastery	–	−.88	.16	−.52	–	−5.59***	
Conscientiousness		–	.39	14	.24	–	2.83**	
E, N, C		.50	–	–	–	16.41***	–	
Extraversion	Self-acceptance	–	.71	.27	.28	–	2.61**	
Neuroticism		–	−.90	.22	−.44	–	−4.04***	
E, N		.32	–	–	–	8.10***	–	
Extraversion		–	−.44	.20	−.24	–	−2.15*	
Neuroticism	Autonomy	–	−.73	.71	−5.1	–	−4.40***	
E, N		.22	–	–	–	5.45***	–	
Conscientiousness		–	.46	.16	.29	–	2.82**	
Openness (O)	Personal growth	–	.97	.25	.46	–	3.94***	
C, O		.23	—	—	–	5.70***	–	
Conscientiousness	Purpose in life	.24	.74	.16	.49	5.81	4.76***	
								
Psychological well-being	
Self-acceptance	Life satisfaction	.52	.67	.11	.56	18.21	6.24***	
Positive relations (PR)		–	−.21	.08	−.27	–	−2.59**	
	–	.40	.08	.51	–	4.83***	
Environmental mastery (EM)	Positive affect	–	.16	.07	.24	–	2.19*	
Self-acceptance (SA)		.30	–	–	–	18.20	–	
PR, EM, SA								
Environmental mastery	Negative affect	.21	−.36	.09	−.48	5.06	−4.24***	
Notes.

* p < .05.

** p < .01.

*** p < .001.

Each of the six dimensions of psychological well-being were related to at least one of the subjective well-being measures: life satisfaction was positively related to self-acceptance, positive affect was positively related to environmental mastery and self-acceptance but negatively related to positive relations with others, and negative affect was negatively related to environmental mastery. See Table 2 for details.

Iranian adolescents:Table 4 displays the correlations between personality and well-being variables for the Iranian sample. In accord with the Swedish sample and our hypothesis, the MRA for the Iranian data showed that Neuroticism was negatively related to life satisfaction and positively related to negative affect. In contrast to the results among Swedes, Neuroticism was not associated to positive affect and Extraversion was not related to any subjective well-being measure (see Table 5). Instead, Openness was positively related to life satisfaction and positive affect. The only psychological well-being dimensions related to personality traits among Iranian adolescents were personal growth (positively related to Extraversion) and purpose in life (positively related to Openness). Also, in contrast to the results among Swedish adolescents, only self-acceptance was positively related to life satisfaction and positive affect. Negative affect was not associated to any of the six psychological well-being dimensions. See Table 5 for details.

Table 4 Correlations between personality and well-being variables in the Iranian sample.

	1	2	3	4	5	6	7	8	9	10	11	12	13	14	15	
Extraversion (1)	–															
Neuroticism (2)	−.09ns	–														
Agreeableness (3)	.11ns	.10ns	–													
Conscientiousness (4)	.18*	.32***	.08ns	–												
Openness (5)	.43***	.15ns	.04ns	.38***	–											
Life satisfaction (6)	.18*	−.19*	−.01ns	.17ns	.26**	–										
Positive affect (7)	.26**	−.01ns	.10ns	−.02ns	.25**	.08ns	–									
Negative affect (8)	.07ns	.24**	.01ns	.19*	.18*	−.07ns	−.03ns	–								
Psychological well-being (9)	.11ns	.03ns	−.07ns	.11ns	.16ns	.19*	.18*	−.18*	–							
Positive relations (10)	−.12ns	.11ns	−.03ns	−.05ns	−.00ns	−.08ns	.03ns	-.18*	.53***	–						
Environmental mastery (11)	−.03ns	−.13ns	−.11ns	.07ns	−.02ns	.09ns	−.02ns	.00ns	.56***	.01ns	–					
Self-acceptance (12)	.21**	−.10ns	−.07ns	.08	.26**	.38***	.31***	−.18ns	.54***	.11ns	.15ns	–				
Autonomy (13)	.20*	.15ns	.14	.19*	.11ns	.08ns	.10ns	−.12ns	.50***	.00ns	.20*	.16ns	–			
Personal Growth (14)	.18*	.08ns	−.05ns	.13ns	.04ns	−.06ns	.05ns	.15ns	.49***	.09ns	.17ns	.10ns	.14ns	–		
Purpose in life (15)	.11ns	−.02ns	−.10ns	−.04ns	.20*	.22**	.13ns	−.25**	.63***	.30***	.23**	.29***	.14ns	.03ns	–	
Notes.

ns nonsignificant

* p < .05.

** p < .01.

*** p < .001.

Table 5 Multiple Regressions for the Iranian sample.

Personality traits’ relationship to subjective well-being and psychological well-being, as well as the relationship between psychological well-being dimensions and subjective well-being.

Predictor
variable	Outcome
variable	Adj R2	Unst. B	Unst. SE	Stand. β	F	t	
Personality traits	
	
Neuroticism (N)		–	−.80	.30	−.26	–	−2.72**	
Openness (O)	Life satisfaction	–	.55	.28	.21	–	1.98*	
N, O		.14	–	–	–	3.44**	–	
Openness	Positive affect	.13	.26	.13	.21	2.74	1.97*	
Neuroticism	Negative affect	.10	.24	.11	.20	2.14	2.07*	
Extraversion	Personal growth	.07	.41	.20	.22	1.30	2.03*	
Openness	Purpose in life	.06	.37	.19	.23	1.43	2.02*	
								
Psychological well-being	
Self-acceptance	Life satisfaction	.16	.56	.18	.32	2.87	3.22**	
Self-acceptance	Positive affect	.13	.26	.09	.30	2.29	3.02**	
Notes.

* p < .05.

** p < .01.

Mediation analyses between personality traits, subjective well-being and psychological well-being

We conducted mediation analyses applying procedures recommended by Baron & Kenny (1986) to test if the effect of personality on subjective well-being was mediated by psychological well-being. For this specific analysis we created a subjective well-being total score by first standardizing positive affect, negative affect, and life satisfaction, and then subtracting the standardized negative affect score from the standardized positive affect score and finally summarizing life satisfaction (i.e., (zpositive affect − znegative affect) + zlife satisfaction). Personality variables were set to be the predictors, the psychological well-being total score as the mediator and subjective well-being total score as the outcome variable. Importantly, as recommended by Baron & Kenny (1986), only personality traits that significantly predicted the outcome variable (i.e., subjective well-being) and the mediator (i.e., psychological well-being) were used in the analyses. For the Swedish sample, Extraversion and Neuroticism met these criteria. However, although Openness and Neuroticism predicted subjective well-being in the Iranian sample, neither of the traits predicted psychological well-being in this sample. Hence, mediation analyses could only be conducted for the Swedish sample.

A series of equations were conducted for analyzing the associations between personality (predictor variable), psychological well-being (mediator), and subjective well-being (outcome variable) within the Swedish dataset. Firstly, psychological well-being was regressed on Extraversion and Neuroticism. Both Extraversion (positively: β = .28, t = 3.30, p < .001) and Neuroticism (negatively: β = −.53, t = −6.17, p < .001) were related to psychological well-being. Secondly, applying hierarchal regression, subjective well-being was regressed on Extraversion and Neuroticism in the first step and to psychological well-being in the second step. In the first step Extraversion (β = .26, t = 2.82, p < .01) and Neuroticism (β = −.51, t = −5.40, p < .001) were significantly associated with subjective well-being. In the second step, the effect of Extraversion on subjective well-being was no longer significant (β = −.18, t = 1.90, p = .06) while Neuroticism was still associated with subjective well-being (β = −.33, t = −3.01, p = .004). Moreover, in the second step of the regression, psychological well-being was associated with subjective well-being (β = .33, t = 2.94, p = .004) and even increasing the prediction of the model (ΔR2 = .06, F(1, 75) = 8.62, p = .004). Psychological well-being yielded a Sobel Z-value = 1.80 (p = .07) for the relationship Extraversion-psychological well-being-subjective well-being and a Sobel Z-value = 2.08 (p < .05) for the relationship Neuroticism-psychological well-being-subjective well-being (see Fig. 1). In other words, these reductions in beta weights suggest that psychological well-being serves to partially mediate the relationship between Extraversion and subjective well-being and the relationship between Neuroticism and subjective well-being.

Figure 1 Model of the mediating role of psychological well-being in the relationships between Neuroticism and subjective well-being and between Extraversion and subjective well-being among Swedish adolescents.

Values in parentheses are the reduced beta weights when the mediator (i.e., psychological well-being) is present. Note: ∗p < .05, ∗∗p < .01, ∗∗∗p < .001, ns = non-significant.

Discussion

The current study investigated variations in well-being in the context of cross-cultural personality traits among adolescents. In accordance with previous studies (e.g., Veenhoven, 2013; Diener & Tov, 2009), Swedish participants reported higher overall well-being than Iranian participants. Furthermore, personality profiles differed significantly between cultures. In accordance with other studies among adults (e.g., Allik & McCrae, 2004), Iranian adolescents reported higher Agreeableness than Swedish adolescents. Further, the Iranian adolescents also reported higher Conscientiousness and Openness than Swedish adolescents. The cultures, however, did not differ significantly in Extraversion and Neuroticism; although, notably Swedish boys reported significantly lowest level of Neuroticism and Swedish girls reported significantly higher levels of Neuroticism than all boys.

Importantly, the present analyses also indicated that cultural variations in self-reported personality traits appear to, in part, be related to differences in well-being. The analyses revealed that in the Swedish sample, Extraversion was positively related to most measures of well-being, and that Conscientiousness was positively related to three of the six psychological well-being dimensions: environmental mastery, personal growth, and purpose in life. This is in accordance with research illustrating that adolescents who are goal-directed and self-controlled (or showing a high degree of agency, which the three psychological well-being dimensions might be argued to tap into; Schütz et al., 2013) are found to also report high levels of well-being (Garcia, 2011a; Garcia, 2012b). In the Iranian sample, in contrast, Openness (rather than Extraversion) was positively related to three well-being measures: life satisfaction, positive affect, and purpose in life.

Interestingly, the Iranian adolescents scored significantly higher on Openness and it was, only for them, positively associated with well-being. At the same time, Swedish and Iranian adolescents did not differ in Extraversion whilst this trait was only positively associated with well-being within the Swedish sample. Conscientiousness was higher represented in Iran but related to well-being within the Swedish sample. Hence, this partially supports our hypothesis, demonstrating that different personality traits can have different influences on well-being in different cultures. However, with the exception of Openness among Iranians, these associations seem to not depend on whether specific personality traits are salient in the culture.

In terms of notable similarities between the two cultures, the psychological well-being dimension of self-acceptance was positively related to high subjective well-being in both samples. This specific result is in accordance to earlier findings regarding adolescents’ subjective well-being (e.g., Garcia & Siddiqui, 2009b; Garcia, 2011b; Garcia, 2012a). That is, adolescents who are satisfied with their lives and experience more positive than negative affect seem to be accepting of all parts of their personality. This specific attitude to the self might be adaptive because it allows the individual to be aware of the self without judging some characteristics as negative or positive; non-evaluative self-awareness is indeed an important factor for well-being (Cloninger, 2004; Cloninger, 2006; Cloninger & Zohar, 2011; Garcia, Anckarsäter & Lundström, 2013; Garcia et al., 2013b; Garcia, Nima & Archer, 2013).

Further, Neuroticism was related to a wide range of well-being dimensions in both cultures; in the Swedish sample it was associated with all subjective well-being components (i.e., negatively related to life satisfaction and positive affect, and positively related to negative affect) and psychological well-being (i.e., negatively related to psychological well-being total score, positive relations, environmental mastery, self-acceptance, and autonomy). Similarly in the Iranian sample, with the exception of positive affect, Neuroticism was negatively related to life satisfaction and positively related to negative affect. Altogether, the strong relationship between Neuroticism and subjective well-being lends important support for the high relevance of Neuroticism put forward by Vittersø (2001), among others. However, Neuroticism was not significantly associated with psychological well-being in the Iranian sample. In addition, as earlier described, a difference between the samples is that Swedish boys reported the significantly lowest level of Neuroticism of all groups, and Swedish girls reported significantly higher levels of Neuroticism than all boys. Hence this gender difference across cultures might be worth scrutinizing in future research.

Limitations

This study was based on self-reported data and correlational methods that ought to be replicated and supported with other methods such as experimental studies. Further, considering the rather small size of both samples, cautiousness in generalizing the results is warranted as well as acknowledging the need for replicating the core purposes of the study. Nevertheless, with the exception of the differences between the samples in purpose in life and psychological well-being total score, the Observed Power was between .84–1.00—an Observed Power of .80 is generally considered acceptable (Cohen, 1988). It is also important to note that the use of current well-being measures can be seen to mirror Western cultures’ conceptions of well-being and might not necessarily be equally suited for Eastern Muslim samples (Joshanloo, 2012). Perhaps this explains the substantial difference in reliability of used scales between the two samples. To address this in future research might involve a re-conceptualization of well-being and the development of new culturally sensitive scales (Joshanloo, 2012). Nonetheless, these results make up an initial platform to build further research upon.

Unfortunately the Iranian dataset did not allow for the mediation analyses because the statistical criteria for such analyses put forward by Baron & Kenny (1986) were not met. Perhaps this is related to the cultural differences in the conceptions of well-being discussed above. Nevertheless, within the Swedish sample, mediation analyses revealed that psychological well-being mediated the relationship between both Neuroticism and Extraversion, to subjective well-being. Hence, this lends empirical support that psychological well-being promote subjective well-being as theorized by Ryan & Deci (2001), among others.

Conclusions

The current study is an addition to previous research showing that Swedes overall tend to report higher well-being than Iranians. Certain personality traits such as Extraversion, Openness, and Conscientiousness relate differently to subjective well-being and psychological well-being among adolescents in these two cultures. Meanwhile, Neuroticism seems to relate similarly across cultures; at least with regard to subjective well-being. Furthermore, the results give an indication on how psychological well-being might mediate the relationship between certain personality traits and subjective well-being. Overall, to achieve a comprehensive understanding of the relationship between personality and well-being, the current results support the importance of contextualizing well-being research at the same time as also employing various well-being measures.

“Personality goes a long way”

Jules in Pulp Fiction

The authors are indebted to the participants for their help in facilitating the study and to Patricia Rosenberg, Saleh Moradi, Reza Molaei, Hasan Kiaei, and Azar Taheri for assistance with the Swedish and Iranian data.

Additional Information and Declarations

Competing Interests

Author Contributions

The authors declare they have no competing interests.

Oscar N.E. Kjell, Trevor Archer and Sverker Sikström wrote the paper.

Ali A. Nima performed the experiments, analyzed the data, contributed reagents/materials/analysis tools, wrote the paper.

Danilo Garcia conceived and designed the experiments, performed the experiments, analyzed the data, contributed reagents/materials/analysis tools, wrote the paper.

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
