# Peer review of "Iranian and Swedish adolescents: differences in personality traits and well-being"

_PeerJ, doi:10.7717/peerj.197_

## Round 0.1 · original submission · Minor Revisions

Overall, this is a well conducted study which is appropriately analysed. The reviewers have carefully scrutinised the paper, as have I, and there remain a number of issues which, if addressed, would improve the paper. I draw the authors’ attention to the entirety of each review, but specifically, would appreciate it if the following issues could be considered prior to a resubmission:

• Please consider reviewers’ comments concerning the formatting and reporting of statistics.
• Sampling issues need to be addressed – how were schools/participants chosen?
• The extent of the generalisability of the findings from a reasonable small sample needs to be considered as a potential limitation.
• Also related to sample size: Either a formal power analysis, or another justification of the sample size as being appropriate, should be included.
• Please address one reviewer’s concerns around the theoretical distinctions between the Big Five (Lexical model) vs Five Factor Model.
• It is also important to include in this paper details of good ethical practice. Please indicate how informed consent, right to withdraw, protection from harm, etc. were ensured, and further indicate where this was scrutinised prior to data collection. This is particularly important given the age of the samples.
• Finally, it is desirable that the data for this study are made available in a publicly accessible archive – for example, http://data-archive.ac.uk/ in the UK, although many other alternatives exist.

·

Basic reporting

This is a well-written article in which the authors have reviewed relevant literature systematically before defining their own hypotheses. The findings of this study will contribute to cross-cultural (comparative) studies among adolescents in general and well-being in particular.

There are some minor issues which I hope the authors consider in their revision. The first two issues relating to ‘basic reporting’ are as follows:

The statistics reported in the final column (far right hand) in Tables 1, 2, and 4 falls outside of margin in the printed copy. They need to be aligned so that they are clearly visible in printed version of the paper.

Although the authors have described in detail the findings of multiple regression analysis (Table 3 and Table 5) in results section (from lines 217 to 231 for the Swedish sample and then from lines 234 to 243 for the Iranian sample) by highlighting the variables which revealed statistical significance, it might be convenient for readers if authors report statistical significance using * mark at the bottom of these two tables (like the way it is done for Table 2 and Table 4).

Experimental design

In the method section, the authors have reported total number of respondents (including information on gender) for both samples. They have also mentioned that respondents are from one high school in Sweden and two high schools from Iran. However, they did not mention how those schools (including classrooms and students within class) were selected. Is it done using purposive or simple random sampling? I think this information is vital for the method section. Authors can add a few sentences explaining the sampling in the ‘Procedure’ sub-section under Method section.

Validity of the findings

Although the authors have drawn their conclusions by analysing data with sophisticated statistical techniques, this study is based on a small sample. There is always a risk of generalising findings obtained from a small study over a large cultural context (Sweden and Iran in this case). The risk also increases if non-probability sampling is used. I think the authors need to mention this as one of the limitations of their study (under Limitations section). Acknowledging this point might bring more credibility of their research to the reader.

Additional comments

I have enjoyed reading the paper. I hope the authors find these comments useful.

Reviewer 2 ·

Basic reporting

The paper aims at investigating the relationship between the broadest five personality domains and two different constructs of human well-being in two samples of adolescents. I believe this is an important issue since it is a relevant contribution to the study of the impact of the personality domains in the two wellbeing measures simultaneously. Furthermore, the existence of two culturally different samples (Iranian and Swedish), can bring significant clarifications to understanding the relationship between personality and well-being across cultures.

Experimental design

First, the literature review is clear, but I think that the authors should integrate summary ideas related to the theoretical conceptualization of the five personality domains and SWB and PWB. Additionally, concerning personality measures, the authors should distinguish the Big Five (lexical tradition) and Five Factor Model; in the present study they use The Big Five Inventory. Although research has revealed relevant similitude between these two measures of personality, they are not the same thing. Moreover, it is important to explain the most important differences between the two well-being concepts.
2. It seems to me that the authors conducted the adequate statistical procedures, although in the presentation a distinction between descriptive and inferential statistics would allow a better reading of the data. The use of Baron and Kenny mediation model is one of the possible procedures and is acceptable. Sobel test was conducted. Concerning figure 1 and the graphics of the results, if the authors used the AMOS program these graphical presentations would be automatic.
3. As the authors stated, the mediation analyses was not conducted in the Iranian sample since the number of subjects was not enough. Because of this we don’t ‘know if the mediation effect, an important aspect of the study, is similar to that of the Swedish sample.

Validity of the findings

No comments.

---

## Round 0.2 · Minor Revisions

Many thanks for addressing points raised in the previous request for minor revisions, and the manuscript is now very close to being ready to accept. I would be grateful if you could address inconsistencies in the referencing (use of DOIs). There are no other issues outstanding.

---

## Round 0.3 · accepted · Accept

Thanks for this resubmission. I am happy to recommend this for publication at this stage. I look forward to reading more of your work in this area in the future.